



# A Fortran-Python Interface for Integrating Machine Learning Parameterization into
Earth System Models
Tao Zhang[1], Cyril Morcrette[2,7], Meng Zhang[3], Wuyin Lin[1], Shaocheng Xie[3], Ye Liu[4], Kwinten Van
Weverberg[5,6], Joana Rodrigues[2]
1. Brookhaven National Laboratory, Upton, NY, USA
2. Met Office, FitzRoy Road, Exeter, EX13PB, UK
3. Lawrence Livermore National Laboratory, Livermore, CA, USA
4. Pacific Northwest National Laboratory, Richland, WA, USA
5. Department of Geography, Ghent University, Belgium
6. Royal Meteorological Institute of Belgium, Brussels, Belgium
7. Department of Mathematics and Statistics, Exeter University, Exeter, UK
Correspondence to: Tao Zhang (taozhang.ccs@gmail.com)
## Abstract
Parameterizations in Earth System Models (ESMs) are subject to biases and uncertainties arising from
subjective empirical assumptions and incomplete understanding of the underlying physical processes.
Recently, the growing representational capability of machine learning (ML) in solving complex problems
has spawned immense interests in climate science applications. Specifically, ML-based parameterizations
have been developed to represent convection, radiation and microphysics processes in ESMs by learning
from observations or high-resolution simulations, which have the potential to improve the accuracies and
alleviate the uncertainties. Previous works have developed some surrogate models for these processes
using ML.  These surrogate models need to be coupled with the dynamical core of ESMs to investigate
the effectiveness and their performance in a coupled system. In this study, we present a novel Fortran-
Python interface designed to seamlessly integrate ML parameterizations into ESMs. This interface
showcases high versatility by supporting popular ML frameworks like PyTorch, TensorFlow, and Scikit-
learn. We demonstrate the interface's modularity and reusability through two cases: a ML trigger function
for convection parameterization and a ML wildfire model. We conduct a comprehensive evaluation of
memory usage and computational overhead resulting from the integration of Python codes into the



Fortran ESMs. By leveraging this flexible interface, ML parameterizations can be effectively developed,
tested, and integrated into ESMs.

## Plain Language

Earth System Models (ESMs) are crucial for understanding and predicting climate change. However, they
struggle to accurately simulate the climate due to uncertainties associated with parameterizing sub-grid
physics. Although higher-resolution models can reduce some uncertainties, they require significant
computational resources. Machine learning (ML) algorithms offer a solution by learning the important
relationships and features from high-resolution models. These ML algorithms can then be used to develop
parameterizations for coarser-resolution models, reducing computational and memory costs. To
incorporate ML parameterizations into ESMs, we develop a Fortran-Python interface that allows for
calling Python functions within Fortran-based ESMs. Through two case studies, this interface
demonstrates its feasibility, modularity and effectiveness.

## 1. Introduction

Earth System Models (ESMs) play a crucial role in understanding the mechanism of the climate system
and projecting future changes. However, uncertainties arising from parameterizations of sub-grid
processes pose challenges to the reliability of model simulations (Hourdin et al., 2017).  Kilometer-scale
high-resolution models (Schär et al., 2020) can potentially mitigate the uncertainties by directly resolving
some key subgrid-scale processes that need to be parameterized in conventional low-resolution ESMs.
Another promising method, superparameterization – a type of multi-model framework (MMF) (D.
Randall et al., 2003; D. A. Randall, 2013), explicitly resolves sub-grid processes by embedding high-
resolution cloud-resolved models within the grid of low-resolution models. Consequently, both high-
resolution models and superparameterization approaches have shown promise in improving the
representation of cloud formation and precipitation. However, their implementation is challenged by
exceedingly high computational costs.

In recent years, machine learning (ML) techniques have emerged as a promising approach to
improve parameterizations in ESMs. They are capable of learning complex patterns and
relationships directly from observational data or high-resolution simulations, enabling the
capture of nonlinearities and intricate interactions that may be challenging to represent with



traditional parameterizations. For example, Zhang et al. (2021) proposed a ML trigger function
for a deep convection parameterization by learning from field observations, demonstrating its
superior accuracy compared to traditional CAPE-based trigger functions. Chen et al. (2023)
developed a neural network-based cloud fraction parameterization, better predicting both spatial
distribution and vertical structure of cloud fraction when compared to the traditional Xu-Randall
scheme (Xu & Randall, 1996). Krasnopolsky et al. (2013) prototyped a system using a neural
network to learn the convective temperature and moisture tendencies from cloud-resolving
model (CRM) simulations. These tendencies refer to the rates of change of various atmospheric
variables over one time step, diagnosed from particular parameterization schemes. These studies
lay the groundwork for integrating ML-based parameterization into ESMs.

However, the aforementioned studies primarily focus on offline ML of parameterizations that do
not directly interact with ESMs. Recently, there have been efforts to implement ML
parameterizations that can be directly coupled with ESMs. Several studies have developed ML
parameterizations in ESMs by hard coding custom neural network modules, such as O'Gorman
& Dwyer (2018), Rasp et al. (2018), Han et al. (2020) and Gettelman et al. (2021). They
incorporated a Fortran-based ML inference module to allow the loading of the pre-trained ML
weights to reconstruct the ML algorithm in ESMs. The hard-coding has limitations. Kochkov et
al. (2023) presented an innovative ML parameterization that feeds back from the dynamics, in
order to improve stability and reduce bias. However, such hard-coding approach restricts the ML
algorithm's ability to adapt to changes in the model dynamics over time, as the 'online' updating
requires a two-way coupling between the dominantly Fortran-based  ESMs and Python ML
libraries.

Fortran-Keras Bridge (FKB; Ott et al. (2020)) and C Foreign Function Interface (CFFI;
https://cffi.readthedocs.io) are two packages that support two-way coupling between Fortran-based ESM
and Python based ML parameterizations. FKB enables tight integration of Keras deep learning models but
is specifically bound to the Keras library, limiting its compatibility with other frameworks like PyTorch
and Scikit-Learn. On the other hand, CFFI provides a more flexible solution that in principle supports
coupling various ML packages due to its language-agnostic design. Brenowitz & Bretherton (2018)
utilized it to enable the calling of Python ML algorithms within ESMs. However, the CFFI has several
limitations. When utilizing CFFI to interface Fortran and Python, it uses global data structures to pass





variables between the two languages. This approach results in additional memory overhead as variable
values need to be copied between languages, instead of being passed by reference. Additionally, CFFI
lacks automatic garbage collection for the unused memory within these data structures and copies.
Consequently, the memory usage of the program gradually increases over its lifetime. In addition, when
using CFFI to call Python functions from a Fortran program, the process involves several steps such as
registering variables into a global data structure, calling the Python function, and retrieving the calculated
result. These multiple steps can introduce computational overhead due to the additional operations
required.

Additionally, Wang et al. (2022) developed a coupler to facilitate two-way communication between ML
parameterizations and host ESMs. The coupler gathers state variables from the ESM using the Message
Passing Interface (MPI) and transfers them to a Python-based ML module. It then receives the output
from the Python code and returns them to the ESM. While this approach effectively bridges Fortran and
Python, its use of file-based data passing to exchange information between modules carries some
performance overhead relative to tighter coupling techniques. Optimizing the data transfer, such as via
shared memory, remains an area for improvement to fully leverage this coupler's ability to integrate
online-adaptive ML parameterizations within large-scale ESM simulations, which is the main goal for this
study.

In this study, we investigate the integration of ML parameterizations into Fortran-based ESM
models by establishing a flexible interface that enables the invocation of ML algorithms in
Python from Fortran. This integration offers access to a diverse range of ML frameworks,
including PyTorch, TensorFlow, and Scikit-learn, which can effectively be utilized for
parameterizing intricate atmospheric and other climate system processes. The coupling of the
Fortran model and the Python ML code needs to be performed for thousands of model columns
and over thousands of timesteps for a typical model simulation. Therefore, it is crucial for the
coupling interface to be both robust and efficient.  We showcase the feasibility and benefits of
this approach through case studies that involve the parameterization of deep convection and
wildfire processes in ESMs. The two cases demonstrate the robustness and efficiency of the
coupling interface. The focus of this paper is on documenting the coupling between the Fortran
ESM and the ML algorithms and systematically evaluating the computational efficiency and
memory usage of different ML frameworks (such as Pytorch and TensorFlow), different ML
algorithms, and different configuration of a climate model. The assessment of the scientific



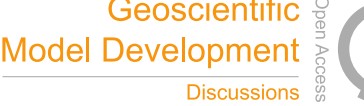

performance of the ML emulators will be addressed in follow-on papers. The showcase examples
emphasize the potential for high modularity and reusability by separating the ML components
into Python modules. This modular design facilitates independent development and testing of
ML-based parameterizations by researchers. It enables easier code maintenance, updates, and the
adoption of state-of-the-art ML techniques without disrupting the existing Fortran infrastructure.
Ultimately, this advancement will contribute to enhanced predictions and a deeper
comprehension of the evolving climate of our planet.

The rest of this manuscript is organized as follows: Section 2 presents the detailed interface that
integrates ML into Fortran-based ESM models. Section 3 discusses the performance of the
interface and presents its application in two case studies. Finally, Section 4 provides a summary
of the findings and a discussion of their implications.

## 2. General design of the ML interface

### 2.1 Architecture of the ML interface

We developed an interface using shared memory to enable two-way coupling between Fortran and Python
(Figure 1). The ESM used in the demonstration in Figure 1 is the U.S. Department of Energy (DOE)
Energy Exascale Earth System Model (E3SM; Golaz et al., 2019, 2022). Because Fortran cannot directly
call Python, we utilized C as an intermediary since Fortran can call C functions. This approach leverages
C as a data hub to exchange information without requiring a framework-specific binding like KFB. As a
result, our interface supports invoking any Python-based ML package such as PyTorch, TensorFlow, and
scikit-learn from Fortran. While C can access Python scalar values through the built-in
PyObject_CallObject function from the Python C API, we employed Cython for its ability to transfer
array data between the languages. Using Cython, multidimensional data structures can be efficiently
passed between Fortran and Python modules via C, allowing for flexible training of ML algorithms within
ESMs.

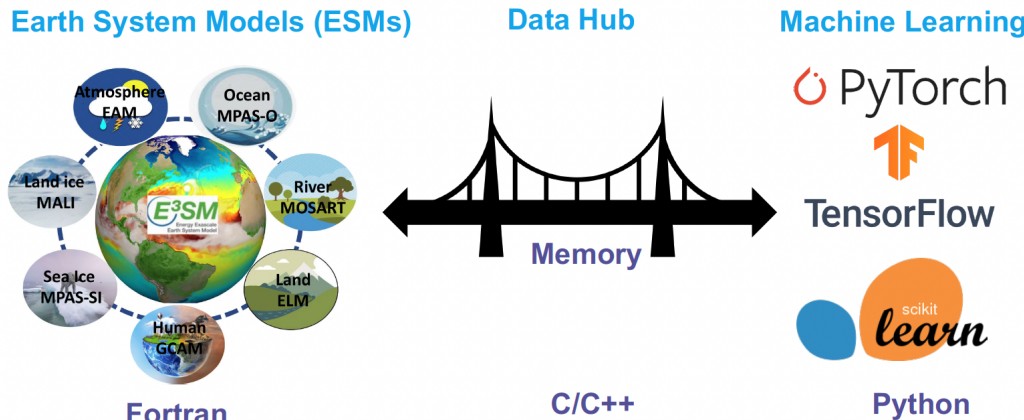

**152**

**153**  **Figure 1.** The interface of the ML bridge for two-way communication via memory between Fortran ESM
**154**  and Python ML module. The diagram for the ESMs uses E3SM as an illustration. Note that MALI and
**155**  GCAM are yet active components of officially released E3SM.

**156**  2.2 Code structure

**157**  Figure 2 illustrates the structure of the ML bridge interface as applied to E3SM. The interface consists of
**158**  four main components: the Fortran ESM, Fortran Interface, C Bridge, and Python ML. The ML functions
**159**  are invoked within the original Fortran ESM parameterization components, such as the atmospheric
**160**  convection and microphysics modules. This process involves transferring the required input variables to
**161**  Python and defining the expected output variables to be returned to the Fortran component. The Fortran
**162**  Interface and C Bridge play a crucial role in establishing the interface between Fortran and Python. They
**163**  facilitate the transfer of variables between Fortran and Python by utilizing memory references. The ML
**164**  function called within the Fortran ESM is defined in the Fortran interface, which is then bound to a
**165**  corresponding C function. This seamless integration enables efficient communication and data exchange
**166**  between the Fortran and Python components. The Python ML component is responsible for handling ML-
**167**  related tasks, such as loading the trained ML algorithm and using it to make predictions. Cython is used
**168**  to simplify the usage and facilitate the transfer and return of arrays. It allows for efficient integration of
**169**  Python code with C libraries, enhancing performance and enabling seamless array operations within the
**170**  ML component.

**171**

**172**  The interface consists of two stages. The first stage involves initializing the ML environment, which
**173**  persists throughout the model simulations. On the Fortran ESM side, the init_ml() function is called in the
**174**  atm_init_mct module. Through the Fortran Interface and C Bridge, the corresponding function in the



Python ML component is invoked. This function loads the ML-related global data and the trained ML
algorithm. This initialization process is performed only once to enhance efficiency and avoid unnecessary
repetition during the simulations. The second stage involves the actual invocation of the ML process. The
example here is an ML-based closure for the deep convection parameterization. We aim to utilize ML to
calculate Convective Available Potential Energy (CAPE) by utilizing an ML emulator based on high-
resolution cloud-resolving model simulations. We call the cape_ml function in the Fortran module
zm_conv, providing temperature, pressure, and humidity as input variables, and defining the returned
CAPE from the ML side. Through the Fortran Interface and C Bridge, these three variables are passed to
the Python ML component. In the Python ML component, the received variables, along with other pre-
loaded global data and the trained ML algorithm, are used to calculate the ML-based CAPE. The
calculated result is then returned to the Fortran ESM. The Fortran ESM utilizes this ML-derived CAPE to
determine how convection will evolve.

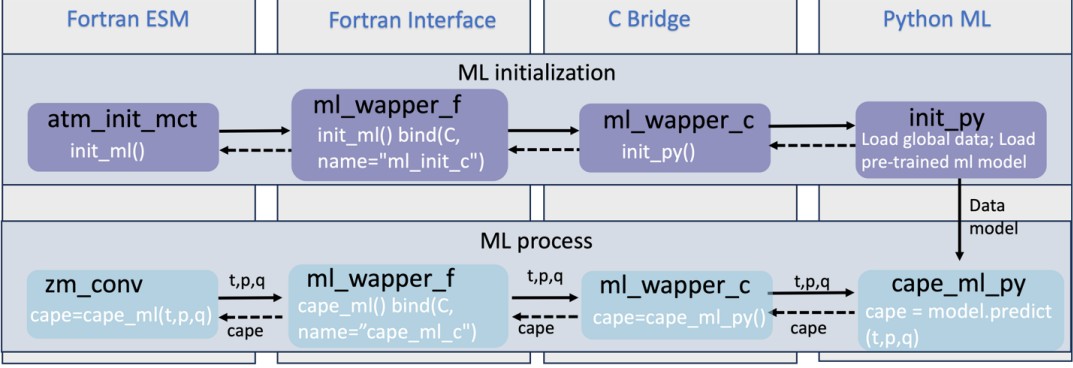

**Figure 2.** The code structure of the ML bridge interface using the ML closure in deep convection as an
example.

In traditional ESMs, sub-grid scale parameterization routines such as convection parameterizations are
often calculated separately for each vertical column of the model domain. Meanwhile, the domain is
typically decomposed horizontally into 2D chunks that can be solved in parallel using MPI processes.
Each CPU core/MPI process is assigned a number of chunks of model columns to update asynchronously
(Figure 3). Our interface takes advantage of this existing parallel decomposition by designing the ML
calls to operate over all columns simultaneously within each chunk, rather than invoking the ML scheme
individually for each column. This allows the coupled model-ML system to leverage parallelism in the
neural network computations. If the ML were called separately for every column, parallel efficiencies



would not be realized. By aggregating inputs over the chunk-scale prior to interfacing with Python,
performance is improved through better utilization of multi-core and GPU-based ML capabilities during
parameterization calculations. The Python, C, Cython and Fortran code components are compiled
together into a unified executable file. Table 1 shows the detailed steps to enable the ML bridge interface
in E3SM.

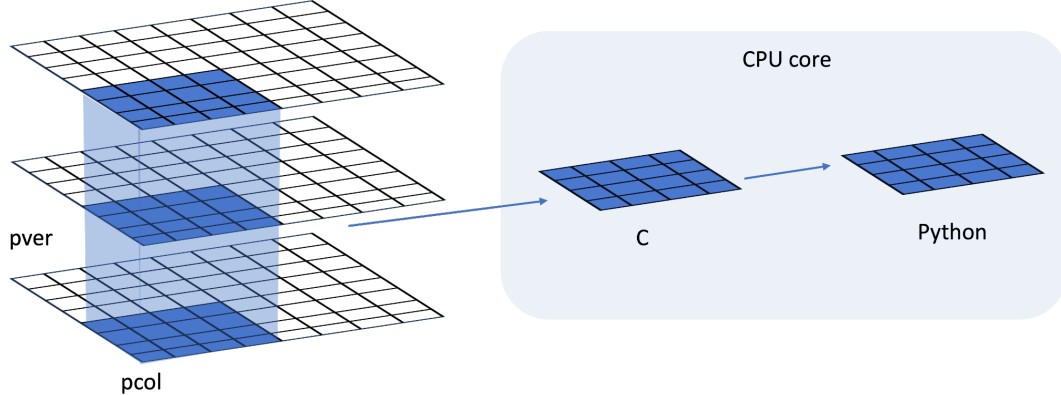


**Figure 3.** Data and system structure. The model domain is decomposed into chunks of columns. pver
refers to number of pressure vertical levels. A chunk contains multiple columns (up to pcol). Multiple
chunks can be assigned to each CPU core.

**Table 1.** The steps to enable the ML bridge framework in E3SM

| Step | Description |
|------|-------------|
| 1. | Create the Python environment using Conda<br>• conda create ML4ESM<br>• conda activate ML4ESM |
| 2. | Add the Python ML environment in the compile CMake file |
| 3. | Incorporate the ML bridge framework codes (including the Fortran Interface and C Bridge) into the ESM codebase. |





4.      Initialize of ML environment by loading necessary global data and the pre-trained ML algorithm.

5.      Implement the ML prediction and the transmission of the resulting values to the ESM parameterization module.

6.      Cythonize the Python code

7.      Build and compile the ESM

8.      Submit the job for model simulation


## 3. Results
The framework explained in the previous section provides seamless support for various ML
parameterizations and various ML frameworks, such as PyTorch, Tensorflow, and Scikit-learn. To
demonstrate the versatility of this framework, we applied it two distinct case applications. The first
application replaces the conventional CAPE-based trigger function in deep convection parameterization
with a machine-learnt trigger function. The second application involves a ML-based wildfire model that
interacts bidirectionally with the ESM. We provide a brief introduction to these two cases. Detailed
descriptions and evaluations will be presented in separate papers.

The framework's performance is influenced by two primary factors: increasing memory usage and
increasing computational overhead. Firstly, maintaining the Python environment fully persistent in
memory throughout model simulations can impact memory usage, especially for large ML algorithms.
This elevated memory footprint increases the risk of leaks or crashes as simulations progress. Secondly,
executing ML components within the Python interpreter inevitably introduces some overhead compared
to the original ESMs. The increased memory requirements and decreased computational efficiency
associated with these considerations can impact the framework's usability, flexibility, and scalability for
different applications.



To comprehensively assess performance, we conducted a systematic evaluation of various ML
frameworks, ML algorithms, and physical models. This evaluation is built upon the foundations
established for evaluating the ML trigger function in the deep convection parameterization.
## 3.1 Application cases
### 3.1.1 ML trigger function in deep convection parameterization
Convection plays a vital role in atmospheric processes, such as precipitation formation, heat and moisture
transport, and energy redistribution (Arakawa, 2004; Arakawa & Schubert, 1974). However, the
deficiencies in convection parameterizations constitute one of the principal sources of uncertainties in
General Circulation Models (D. A. Randall, 2013). Some uncertainties in convection parameterizations
are recognized to be closely linked to the convection trigger function used in these schemes (Bechtold et
al., 2004; Xie et al., 2004, 2019; Xie & Zhang, 2000; Lee et al., 2007). The convective trigger in a
convective parameterization determines when and where model convection should be triggered as the
simulation advances. In many convection parameterizations, the trigger function consists of a simple,
arbitrary threshold for a physical quantity, such as convective available potential energy (CAPE).
Figure 4a illustrates how the CAPE-based trigger function works. Convection will be triggered if the
CAPE value exceeds a threshold value, such as 70 J/kg used in E3SM version 1.

In this work, we develop a ML trigger function and apply it to E3SM (Golaz et al., 2019, 2022). A brief
overview of this ML trigger function is given here, while further details will be elaborated upon in a
subsequent paper. The training data originates from simulations performed using the Met Office Unified
Model Regional Atmosphere 1.0 configuration (Bush et al., 2020). Each simulation consists of a limited
area model (LAM) nested within a global forecast model providing boundary conditions (Walters et al.,
2017; Webster et al., 2008). In total 80 LAM simulations were run located so as to sample different
geographical regions worldwide. Each LAM was run for 1 month, with 2-hourly output, using a grid-
length of 1.5 km, a 512 x 512 domain, and a model physics package used for operational weather
forecasting. This physics package does not include a convective parameterization scheme, but does
include a representation of fractional cloudiness (Bush et al., 2020). The 1.5 km data is coarse-grained to
several scales from 15 to 144 km, comparable to the scale a global model might be run at. At each scale,
we assess whether individual pixels can be considered to be buoyant cloudy updrafts (BCU, e.g.
Hartmann et al., 2019; Swann, 2001). Here, the threshold for buoyant is local virtual temperature more
than 0.1 K warmer than the average at that scale and height. Cloudy is defined whenever the fractional
cloud cover is greater than 0.0 and updraft is defined as vertical ascent larger than 0.2 m/s. In each



averaging region, the number of grid points that meet all three criteria are counted and saved as a profile
of BCU fraction.

A two-stream neural network architecture is used for the ML model. The first stream takes profiles of
temperature, specific humidity and pressure as inputs and passes them through a 4-layer convolutional
neural network (CNN) with kernel sizes of 3, to extract large scale features. The second stream takes
mean orographic height, standard deviation of orographic height, land fraction and the size of the grid-
box as inputs. The outputs of the two streams are then combined and fed into a 2-layer fully connected
network to allow the ML model to leverage both atmospheric and surface features when making its
predictions. The output pf the ML model is a profile of BCU.

Once trained, the CNN is coupled to E3SM and thermodynamic information from E3SM is passed to it to
predict the profile of BCU. If there are 3 contiguous levels where the predicted BCU is larger than 0.05,
the convection scheme is triggered.

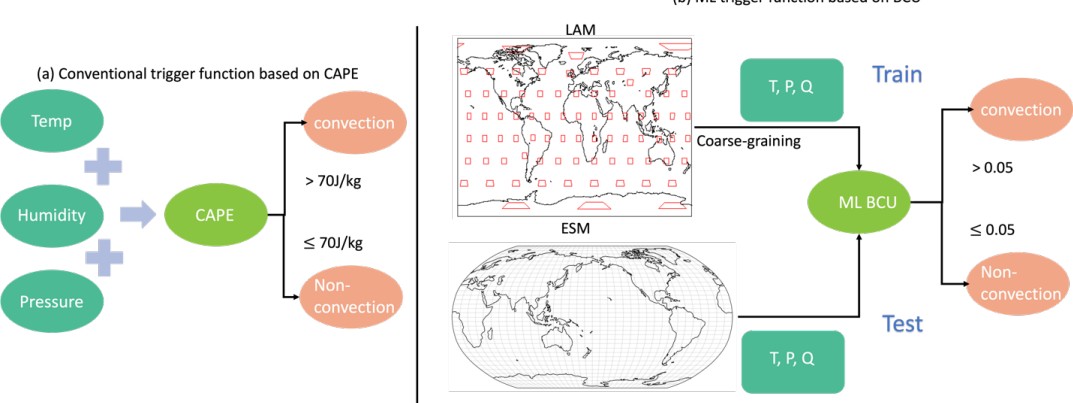


**Figure 4.** Structure of traditional CAPE-based and the new ML BCU-based trigger function. The
rectangles in LAM represent the LAM domains.

The ML trigger function is implemented using this two-stream architecture and coupled with the E3SM
model using the framework described in Section 2. Figure 5 shows the comparison of annual mean
precipitation between the control run using the CAPE-based trigger function and the run using the ML
BCU trigger function. The ML BCU scheme demonstrates reasonable spatial patterns of precipitation,
similar to the control run, with comparable root-mean-square error and spatial correlation. Additional




experiments exploring  the definition of BCU and varying the thresholds along with an in-depth analysis
will be presented in a follow-up paper.




**Figure 5.** Comparison of annual mean precipitation between the control run using the CAPE-based
trigger function (a, c) and the run using the ML BCU trigger function (b, d).

3.1.2 ML learning fire model
Wildfires in the United States have significantly increased in frequency and intensity in recent decades,
resulting in substantial direct and indirect losses (Iglesias et al., 2022). Predicting wildfire burned area is
challenging due to the complex interrelationships between fires, climate, weather, vegetation, topography,
and human activities (Huang et al., 2020). Traditionally, statistical methods like multiple linear regression
have been applied, but are limited in the number and diversity of predictors considered (Yue et al., 2013).
Alternatively, ML algorithms that capture statistical relationships between the burned area and
environmental factors have shown promising burned area prediction (Kondylatos et al., 2022; Li et al.,
2023; Wang et al., 2022, 2023). However, improving long-term burned area projections and evaluating
fire impacts requires the coupling of the fire model to an earth system model, which allows simulations of
the interactions between the fire, atmosphere, land cover and vegetation (Huang et al., 2021). To achieve
this, we develop a coupled fire-land-atmosphere framework using ML.

The ML algorithm is trained using a monthly dataset, which includes the target variable of burned area, as
well as various predictor variables. These predictors encompass local meteorological data (e.g., surface
temperature, precipitation), land surface properties (e.g., monthly mean evapotranspiration and surface



soil moisture), and socioeconomic variables (e.g., gross domestic product, population density), as
described by Wang et al. (2022). In the coupled fire-land-atmosphere framework, meteorology variables
and land surface properties are provided by the E3SM, as illustrated in Figure 6. We use the eXtreme
Gradient Boosting algorithm implemented in Scikit-Learn to train the ML fire model. Figure 7
demonstrates that the ML4Fire model exhibits superior performance in terms of spatial distribution
compared to process-based fire models, particularly in the Southern US region. Detailed analysis will be
presented in a separate paper. The ML4Fire model has proven to be a valuable tool for studying
vegetation-fire interactions, enabling seamless exploration of climate-fire feedbacks.

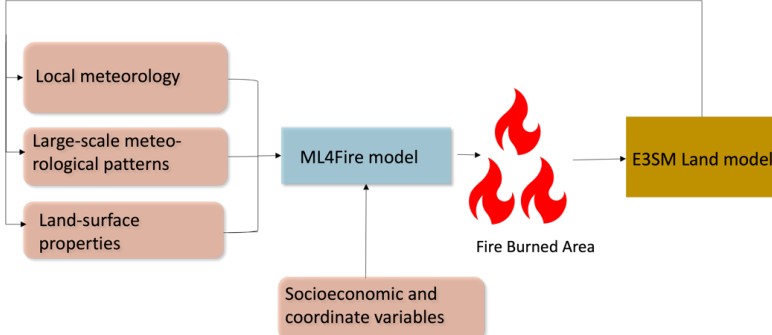

**Figure 6.** Structure of ML fire model (ML4Fire) coupled into E3SM model.

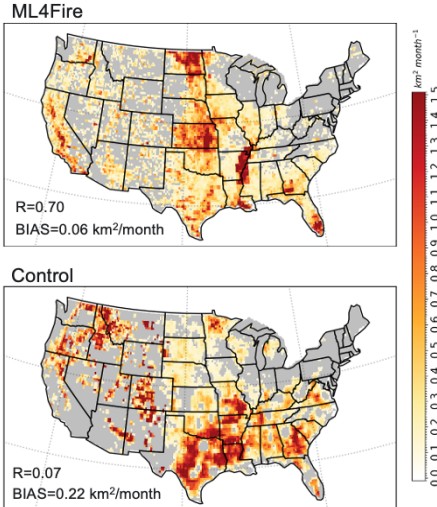


**Figure 7.** Comparison between ML4Fire model and process-based fire model against the historical
burned area from Global Fire Emissions Database 5 from 2001-2020. R and BIAS are the spatial
pattern correlation and difference against the observation, respectively.

## 3.2 Performance of different ML frameworks

The Fortran-Python bridge ML interface supports various ML frameworks, including PyTorch,
TensorFlow, and scikit-learn. These ML frameworks can be trained offline using kilometer-scale high-
resolution models (such as the ML trigger function) or observations (ML fire model). Once trained, they
can be plugged into the ML bridge interface through different API interfaces specific to each framework.
The coupled ML algorithms are persistently resident in memory, just like the other ESM components.
During each step of the process, the performance of the full system is significantly affected by memory
usage. If memory consumption increases substantially, it may lead to memory leaks as the number of time
step iteration increases. In addition, Python, being an interpreted language, is typically considered to have
slower performance compared to compiled languages like C/C++ and Fortran. Therefore, incorporating
Python may decrease computational performance. We examine the memory usage and computational
performance across various ML frameworks based on implementing the ML trigger function in E3SM.
The ML algorithm is implemented as a two-stream CNN model using Pytorch and TensorFlow
frameworks, as well as XGBoost using the Scikit-learn package.

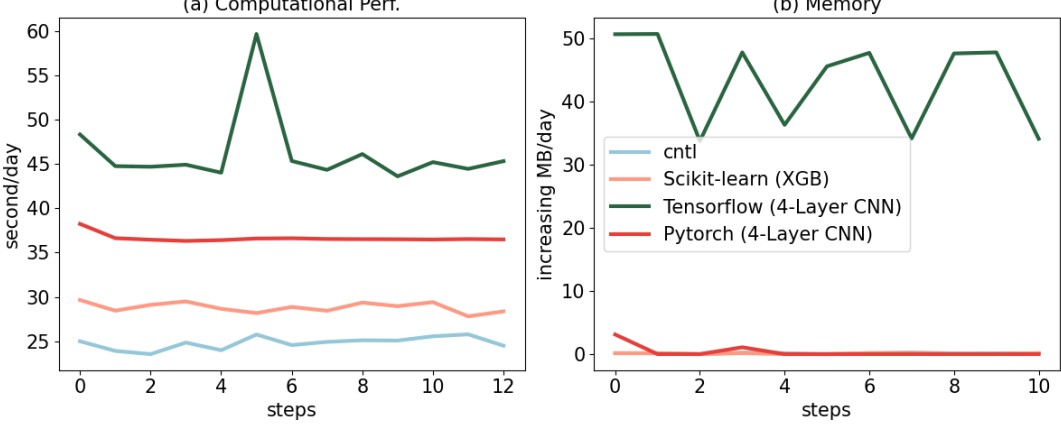


**Figure 8.** Computational and memory overhead as the simulation progresses for coupling the ML trigger
function with the E3SM model. The x-axis represents the simulated time step. The y-axis of (a) represents
the simulation speed measured in seconds per day (indicating the number of seconds required to simulate
one day). The y-axis of (b) represents the relative increase in memory usage for Scikit-learn, TensorFlow,
and PyTorch compared with CNTL. CNTL represents the original simulation without using the ML
framework.

Figure 8 illustrates the computational and memory overhead associated with the ML parameterization
using different ML frameworks. It shows that XGBoost only exhibits a 20% increase in the simulation
time required for simulating one day due to its simpler algorithm. For more complex neural networks,





PyTorch incurs a 52% overhead, while TensorFlow's overhead is almost 100% – about two times as much
as the overhead by PyTorch. In terms of memory usage, we use the highwater memory metric (Gerber &
Wasserman, 2013), which represents the total memory footprint of a process. Scikit-learn and PyTorch do
not show any significant increase in memory usage. However, TensorFlow shows a considerable increase
up to 50MB per simulation day per MPI process element. This is significant because for a node with 48
cores, it would equate to an increase of around 2GB per simulated day on that node. This rapid memory
growth could quickly lead to a simulation crash due to insufficient memory during continuous
integrations, preventing the use in practical simulations. Our findings show that the TensorFlow
prediction function does not release memory after each call. Therefore, we recommend using PyTorch for
complex deep learning algorithms and Scikit-learn for simpler ML algorithms to avoid these potential
memory-related issues when using TensorFlow.

Previous work, such as Brenowitz & Bretherton (2018, 2019) has utilized the CFFI package to establish
communication between Fortran ESM and ML Python. As described in the Introduction, while CFFI
offers flexibility in supporting various ML packages, it does have certain limitations. To pass variables
from Fortran to Python, the approach relies on global data structures to store all variables, including both
the input from Fortran to Python and the output returning to Fortran. Consequently, this package results in
additional memory copy operations and increasing overall memory usage. In contrast, our interface takes
a different approach by utilizing memory references to transfer data between Fortran and Python,
avoiding the need for global data structures and the associated overhead. This allows for a more efficient
data transfer process.

In Figure 9, we present a comparison between the two frameworks by testing the different number of
elements passed from Fortran to Python. The evaluation is based on a demo example that focuses solely
on declaring arrays and transferring them from Fortran to Python, rather than a real E3SM simulation.
Figure 9a illustrates the impact of the number of passing elements on the overhead of the two interfaces.
As the number of elements exceeds $10^4$, the overhead of CFFI becomes significant. When the number
surpasses $10^6$, the overhead of CFFI is nearly ten times greater than that of our interface. Regarding
memory usage, our interface maintains a stable memory footprint of approximately 60MB. Even as the
number of elements increases, the memory usage only shows minimal growth. However, for CFFI, the
memory usage starts at 80MB, which is 33% higher than our interface. As the number of elements
reaches $10^6$, the memory overhead for CFFI dramatically rises to 180MB, twice as much as our interface.





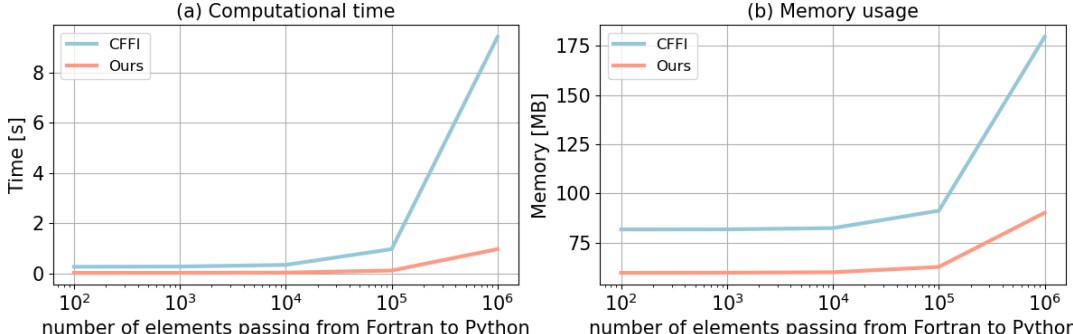


Figure 9. Comparison of our framework and the CFFI framework in terms of computational time
and memory usage. The x-axis represents the number of elements transferred from Fortran to
Python, while the y-axis displays the total time (a) and total memory usage (b) for a
demonstration example. The evaluations presented are based on the average results obtained
from 5 separate tests.

## 3.3 Performance of ML algorithms of different complexities
ML parameterizations can be implemented using various deep learning algorithms with different levels of
complexity. The computational performance and memory usage can be influenced by the complexity of
these algorithms. In the case of the ML trigger function, a two-stream four-layer CNN structure is
employed. We compare this structure with other ML algorithms such as Artificial Neural Network (ANN)
and Residual Network (ResNet), whose structures are detailed in Table 2. These algorithms are
implemented in PyTorch. The algorithm's complexity is measured by the number of parameters, with the
CNN having approximately 60 times more parameters than ANN, and ResNet having roughly 1.5 times
more parameters than CNN.

**Table 2.** The structure and number of parameters of each ML algorithms.

| Algorithms | Structure | # of parameters |
|---|---|---|
| ANN | 3 x Linear | 121,601 |
| CNN | 4 x Conv2d + 2 x Linear | 7,466,753 |
| ResNet | 17 x Conv2d + 1 x Linear | 11,177,025 |




Figure 10 presents a comparison of the memory and computational costs between the CNTL run without
deep learning parameterization and various deep learning algorithms. A same specific process-element
layout (placement of ESM component models on distributed CPU cores) is used for all the simulations.
Deep learning algorithms incur a significant yet affordable increase in memory overhead, with at least a
20% increase compared to the CNTL run (Figure 10a). This is primarily due to the integration of ML
algorithms into the ESM, which persist throughout the simulations. Although there is a notable increase in
complexity among the deep learning algorithms, their memory usage only shows a slight rise. This is
because the memory increment resulting from the ML parameters is relatively small. Specifically, ANN
requires 1MB of memory, CNN requires 60MB, and the ResNet algorithms requires 85MB, which are
calculated based on the number of parameters in each algorithm. When comparing these values to the
memory consumption of the CNTL run, which is approximately 3000MB, the additional parameters'
incremental memory consumption is not substantial.

However, there is a significant decrease in computational performance as the complexity of the deep
learning algorithms increases (Figure 10b). This is primarily due to the larger number of parameters in
neural networks, which require more forward computations. It is worth noting that in this study, the deep
learning algorithms are executed on CPUs. To enhance computational performance, future work could
consider utilizing GPUs for acceleration.

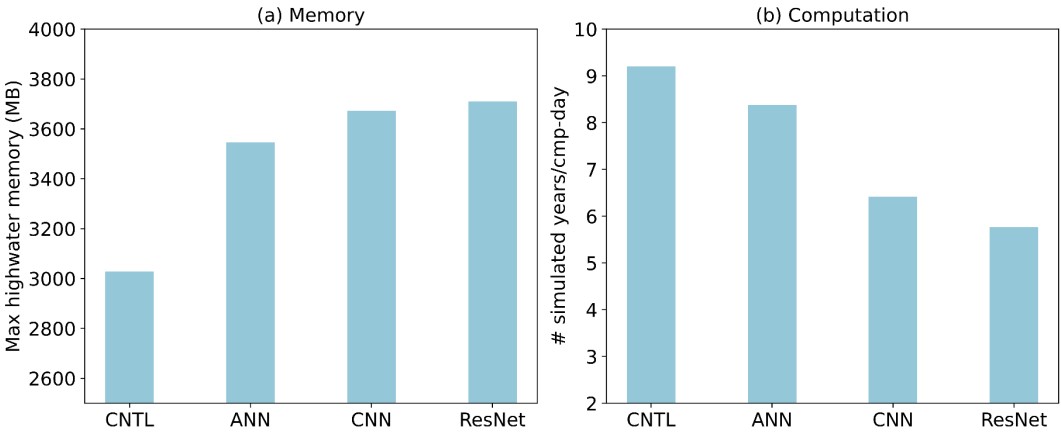


**Figure 10.** Comparison of CNTL and various ML algorithms in terms of memory and computation.
CNTL is the default run without ML parameterizations.





## 3.4 Performance for physical models of different complexities

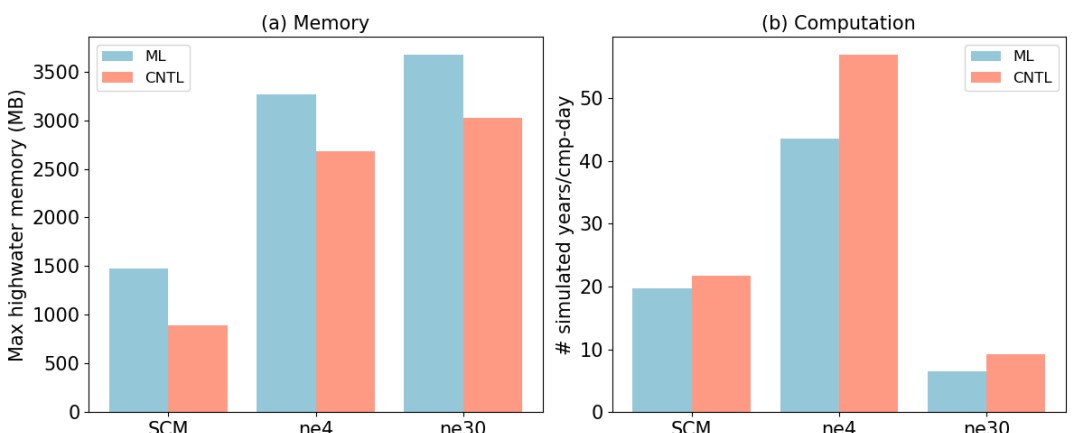

**Figure 11.** Compassion CNTL and ML for various ESMs in terms of memory and computation. The ESM configuration include SCM, ultra-low resolution model (ne4) and nominal low-resolution model (ne30).

ML parameterization can be applied to various ESM configurations, for example, with the E3SM Atmosphere Model (EAM), we experiment with Single Column Model (SCM), the ultra low-resolution model of EAM (ne4), and the nominal low resolution model of EAM (ne30) configurations. The SCM consists of one single atmosphere column of a global EAM (Bogenschutz et al., 2020; Gettelman et al., 2019). ne4 has 384 columns, with each column representing the horizontal resolution of 7.5°. ne30 is the default resolution for EAM and comprises 21,600 columns, with each column representing the horizontal resolution of 1°. In the case of the ML trigger function, the memory overhead is approximately 500MB for all configurations due to the loading of the ML algorithm, which does not vary with the configuration of the ESM.

Regarding computational performance, SCM utilizes 1 process, ne4 employs 1 node with 64 processes, and ne30 utilizes 10 nodes with each node using 128 processes. In the case of SCM, the overhead attributed to the ML parameterization is approximately 9% due to the utilization of only 1 process. However, for ne4 and ne30, the overhead is 23% and 28% respectively (Figure 11). The increasing computational overhead is primarily due to resource competition when multiple processes are used within a single node.





## 4. Discussion and Conclusion

In this study, we develop a novel Fortran-Python interface for developing ML parameterizations. ML
algorithm can learn detailed information about cloud processes and atmospheric dynamics from
kilometer-scale models and observations and serves as an approximate surrogate for the kilometer-scale
model. Instead of explicitly simulating kilometer-scale processes, the ML algorithms can be designed to
capture the essential features and relationships between atmospheric variables by training on available
kilometer-scale data. The trained algorithms can then be used to develop parameterizations for use in
models at coarser resolutions, reducing the computational and memory costs. By using ML
parameterizations, scientists can effectively incorporate the insights gained from kilometer-scale models
for coarser-resolution simulations. Through learning the complex relationships and patterns present in the
high-resolution data, the ML-based parameterizations have the potentials to more accurately represent
cloud processes and atmospheric dynamics in the ESMs. This approach strikes a balance between
computational efficiency and capturing critical processes, enabling more realistic simulations and
predictions while minimizing computational resources. All these potential benefits in turn promote
innovative developments to facilitate increasing and more efficient use of ML parameterizations.

In this study, we develop a novel Fortran-Python interface for developing ML parameterizations. This
interface demonstrates feasibility in supporting various ML frameworks, such as PyTorch, TensorFlow,
and Scikit-learn and enables the effective development of new ML-based parameterizations to explore
ML-based applications in ESMs. Through two cases - a ML trigger function in convection
parameterization and a ML wildfire model - we highlight high modularity and reusability of the
framework. We conduct a systematic evaluation of memory usage and computational overhead from the
integrated Python codes.

Based on our performance evaluation, we observe that coupling ML algorithms using TensorFlow into
ESMs can lead to memory leaks. As a recommendation, we suggest using PyTorch for complex deep
learning algorithms and Scikit-learn for simple ML algorithms for the Fortran-Python ML interface.

The memory overhead primarily arises from loading ML algorithms into ESMs. If the ML algorithms are
implemented using PyTorch or Scikit-learn, the memory usage will not increase significantly. The
computational overhead is influenced by the complexity of the neural network and the number of
processes running on a single node. As the complexity of the neural network increases, more parameters





in the neural network require gradient computation. Similarly, when there are more processes running on
a single node, the integrated Python codes introduces more resource competition.

Although this interface provides a flexible tool for ML parameterizations, it does not currently utilize
GPUs for ML algorithms. In Figure 3, it is shown that each chunk is assigned to a CPU core. However, to
effectively leverage GPUs, it is necessary to gather the variables from multiple chunks and pass them to
the GPUs. Additionally, if an ESM calls the Python ML module multiple times in each time step, the
computational overhead becomes significant. It is crucial to gather the variables and minimize the number
of calls. In the future, we will enhance the framework to support this mechanism, enabling GPU
utilization and overall performance improvement.

## 490    Acknowledge

This work was primarily supported by the Energy Exascale Earth System Model (E3SM) project of the
Earth and Environmental System Modeling program, funded by the US Department of Energy, Office of
Science, Office of Biological and Environmental Research. Research activity at BNL was under the
Brookhaven National Laboratory contract DE-SC0012704 (Tao Zhang, Wuyin Lin). The work at LLNL
was performed under the auspices of the US Department of Energy by the Lawrence Livermore National
Laboratory under Contract DE-AC52-07NA27344. The work at PNNL is performed under the Laboratory
Directed Research and Development Program at the Pacific Northwest National Laboratory. PNNL is
operated by DOE by the Battelle Memorial Institute under contract DE-A05-76RL01830.

## 500    Author contribution

TZ developed the Fortran-Python Interface. CM and JR contributed the ML model for the trigger
function. YL contributed the ML model for the wire fire model. TZ and MZ assessed the performance of
the ML trigger function. TZ took the lead in preparing the manuscript, with valuable edits from CM, MZ,
WL, SX, YL, KW, and JR. All the co-authors provided valuable insights and comments for the
manuscript.

## 506    Conflict of Interest

The authors declare that they have no conflict of interest.



## Data Availability Statement

The Fortran-Python interface for developing ML parameterizations can be archived at
https://doi.org/10.5281/zenodo.11005103 (Zhang et al., 2024). The E3SM model can be accessed at
https://doi.org/10.11578/E3SM/dc.20240301.3 (E3SM Project, 2024).

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
