# Peer review of "A Fortran-Python Interface for Integrating Machine Learning Parameterization into 1 2 Earth System Models 3 Tao Zhang1, Cyril Morcrette2,7, Meng Zhang3, Wuyin Lin1, Shaocheng Xie3, Ye Liu4, Kwinten Van Weverberg5,6, Joan"

_Geoscientific Model Development, 2024_

## Author Comment (AC2)

Summary

My understanding is that this paper's main purpose is to introduce and describe a general approach for coupling Python-based software components to primarily Fortran-based Earth System Models. This is a problem that has become of increasing interest in recent years, with the advent of machine-learning based parameterizations, which are often most convenient to write, train, and evaluate with various different Python frameworks. As the authors note, multiple strategies have been employed for coupling these ML models to ESMs in previous studies. The main contribution of the authors here is to describe an approach which involves writing Cythonized Python functions to carry out initialization and execution of an ML model, C functions which call those Cythonized Python functions as a bridge, and ultimately Fortran functions which bind to those C functions, which can be called from anywhere in the Fortran model. They document the approach some, and then they describe using this framework in real-world scientific applications involving coupling ML parameterizations for a convective trigger function or fire burned area to E3SM, as well as some benchmarking test cases.

I have to admit that I found the scope of this paper to be somewhat sprawling. Documenting this approach for coupling ML models to Fortran models seems valuable, but I think more space and detail could have been devoted to that, with less space devoted to the details of the scientific applications, which the authors note will be described elsewhere. In terms of benchmarks, the direct comparison to the CFFI coupling approach seemed quite relevant, but other aspects like the impact of ML model type and complexity on performance seemed somewhat orthogonal to the choice of coupling method. For example, it does not seem surprising that more complex ML models will be more computationally expensive, regardless of the coupling approach. Maybe there is something I am missing about the motivation that the authors could describe more clearly, but as it stands now, I would like to see improvements to the focus of the manuscript before condsidering recommending publishing.

*Reply: We thank the reviewer for their time in reviewing our paper and useful suggestions, which help to significantly improve our paper! In summary, we enhance the description of interface usage, compare it with existing packages like CFFI to highlight improvements, conduct a systematic evaluation of overheads for ANN, CNN, and ResNet, as these methods are commonly applied in ML parameterizations. Additionally, we reduce the scientific use case descriptions to focus only on inputs, outputs, and ML algorithms based on your suggestion.*

General comments

1. I found Table 1 to be somewhat vague. I feel like a simplified toy code example would go a long way toward illustrating what is required and how everything fits together. To me, for this paper, this is more important than the scientific details of

the case studies, which the authors note will be described more fully in forthcoming papers. There is value in noting that this coupling approach has been successfully used in each of these real-world applications, but I do not think much more needs to be said beyond the general idea of each project, what kind of ML model is used in each, and maybe what the inputs and outputs are. In other words, Figures 4-7, illustrating the structure of these models and skill when they are coupled online, and much of the paragraphs that go along with them, feel outside the scope of this paper.

*Reply: We include toy codes to demonstrate how Fortran calls the Python ML function, passing parameters through memory references and returning results to Fortran, as shown in the following figure. Additionally, we provide a description of how the interface is coupled with the real model.*

*"When coupling the Python ML module with the real model using the interface, additional steps should be considered: 1. The ML module should remain active throughout the model simulations, without any Python finalization calls, ensuring it is continuously available. 2. The Python module should load the trained ML model and any required global data only once, rather than at each simulation step. This one-time initialization process improves efficiency and prevents unnecessary repetition. On the Fortran ESM side, the init_ml() function is called within the atm_init_mct module to load the ML model and global data. Then, similar to the toy code, we call the wrapper function, pass input variables to Python for ML predictions, and return the results to the Fortran side. 3. When compiling the complex system, which includes Python, C, Cython, and Fortran code, the Python path should be specified in the CFLAGS and LDFLAGS. It is important to note that without the position-independent compiling flag (-fPIC) compiling flag, the hybrid system will only work on a single node and may cause segmentation faults on multiple nodes. Including it can resolve this issue, allowing multi-node compatibility."*

*The Table 1 has been removed. The description of the two scientific use cases has been condensed, only providing a brief background, input, output, machine learning algorithm, and key results. Two figures related to the background of the use cases have been removed, only leaving the results.*

[Figure]

*Figure R1. Toy code illustrating the Fortran-Python interface.*

2. What is the intended takeaway of the performance experiments with different types of ML models in Section 3.3? Is this not something that could be learned by profiling the computational performance of the ML models in isolation? There maybe is some value in documenting the relative cost of a typical ML model to a typical climate model simulation, but to some extent one can already get the sense for this through Figure 8(a) or previous ML parameterization papers. In practice the tradeoff will always need to be assessed on a case-by-case basis regarding whether the improvement in hybrid model skill justifies the additional computational cost of the ML model (i.e. this kind of discussion seems better suited for an application-specific paper).

*Reply: Sorry for the confusion. Each bar in Figure 10 represents the performance for the hybrid system that couples the ML methods into the ESM, not just the ML module alone. We have clarified this in the revised text. We selected these three ML algorithms because they are commonly used in previous ML parameterization approaches, (Brenowitz & Bretherton, 2019; Han et al., 2020; Wang et al., 2022). Systematically evaluating the hybrid system with these ML methods using our interface can help identify bottlenecks and improve the system computational performance. Specially, in term of the memory overhead, when we use 128 MPI processes per node, it could bring the total memory requirement to approximately 460 GB per node. If the available hardware memory is less than this, the process layout must be adjusted accordingly. In terms of computational performance, we compare our result with the preivous work. In our work, the performance decrease is not substantial. The simple ANN model reduces performance by only about 10%*

*compared to the CNTL run, while even the more complex ResNet model results in a*
*35% decrease. In contrast, Wang et al. (2022) reported a 100% overhead in their*
*interface, which transfers parameters via files.*

*In addition, we develop a performance model to estimate computational performance*
*for the hybrid model using different ML model sizes and complexities. This*
*performance model, based on linear regression, predicts the computational ratio*
*relative to the CNTL run by taking the number of ML parameters as input, shown in*
*Figure 9b. It provides a simple yet effective way to capture this relationship and*
*serves as a valuable tool for performance prediction when incorporating more*
*complicated ML models. We have included these texts into the revised version.*

[Figure]

*Figure R2.  Comparison of CNTL and the hybrid model using various ML algorithms in*
*terms of memory and computation. CNTL is the default run without ML parameterizations.*
*In (b), the left y-axis represents the actual number of simulated years per day, while the*
*right y-axis shows the relative performance compared to the CNTL run (orange line). The*
*gray line illustrates the regression between the number of ML parameters (x) and the*
*relative performance of the hybrid system (y).*

Specific comments

Lines 79-84: I am not sure I follow the discussion in these lines.  As I understand it, the key
advance of Kochkov et al. (2023) is that their entire model—both the physics-based
dynamics and ML-based physics—is differentiable, enabling feedbacks between the two to
be felt and accounted for in training.  This is more significant than merely enabling greater
flexibility in the ML model one can couple to a Fortran-based GCM.  So long as the GCM is
still written in legacy Fortran I do not think there is anything that can be done to easily
enable differentiation through the entire hybrid model.  In other words, you will still need to

train the ML model in a purely "offline" sense.  A software interface between the ML model and the GCM—however hard-coded or flexible it is—merely enables online testing, which is no doubt important, but not the same as enabling coupling during training.

*Reply: Thanks for the comment. We agree with your point and remove this sentence.*

Lines 115-117: I think it is fair to say that this approach offers access to calling any Python code from Fortran, of which the ML frameworks listed are obviously just a subset.  The phrasing of this line makes it sound as though there is some flexibility, but some frameworks might not be supported.

*Reply: Thanks for the suggestion. We have revised the text by "This integration offers access to any Python codes from Fortran, including a diverse range of ML frameworks, such as PyTorch, TensorFlow, and Scikit-learn, which can effectively be utilized for parameterizing intricate atmospheric and other climate system processes."*

Line 131: "[...] without disrupting the Fortran infrastructure."  This feels maybe a bit overstated—beyond calling the ML code itself within Fortran—which is maybe self-evident—the build system of the now hybrid Fortran/Python model needs to be updated to support these changes, which is not always trivial (e.g. it might be a little easier to build in a bespoke Fortran implementation of an ML model even though that is obviously much less flexible).

*Reply: Your concern is indeed the focus of our work. We aim to minimize the extra effort needed to connect Fortran and Python within the complex GCM software system. Based on your suggestion, we have added example codes to demonstrate the interface and describe how it can be applied to a real GCM system. This allows users to solely focus on the physics and ML methods, without worrying about the interface details.*

*Addressing specific scientific questions often requires exploring various ML methods, as it is unlikely that a single ML model will suit all needs. Additionally, a well-performing offline ML method does not guarantee stable performance in online simulations. Frequent adjustments and improvements to the ML method are necessary. Therefore, it is essential to have a tool to support this flexibility.*

*To avoid overstatement, 'without' has been changed to 'with only minimal'.*

Lines 338-339: if it is to be included here, I think it should be noted that XGBoost is a totally different type of ML model than the CNNs implemented in PyTorch or TensorFlow, so it is

not really an apples-to-apples comparison for computational performance.  This is sort of alluded to in Line 350, but I think it could be made more explicit.

*Reply: Thanks for the suggestion. We have revised the text by "It should be noted that XGBoost, a boosting tree-based model, is a completely different type of ML model compared to the CNNs, which are the type of deep neural network."*

Lines 334-336: as I am sure you are aware, for pure Python, this is true, but most packages designed for numerical computation wrap C/C++ or Fortran.  This is something that is also somewhat orthogonal to the framework one uses for coupling—if the Python code is a bottleneck, it will be a bottleneck no matter how it is coupled.  To truly test the degree to which implementation language was a bottleneck one would need a baseline where the identical ML model was evaluated directly in Fortran (like in Rasp et al., 2018).

*Reply: We agree with your points. If the Python code is the bottleneck, some overhead could be inevitable. However, we could minimize the overhead. In this work, we provide the flexible interface and reduce the overhead by memory reference. In the future work, we will effectively utilize GPUs and leverage specialized Pytorch compilers to reduce the overhead.*

Lines 358-359: do you know if is this a deep fundamental issue with TensorFlow (i.e. hard to fix)?

*Reply:  We tested several methods to manually free TensorFlow memory after calling the predict function, including tf.keras.backend.clear_session() and gc.collect(), but they didn't resolve the issue. According to a discussion on TensorFlow's GitHub page, memory usage persists until the Python process is terminated (https://github.com/tensorflow/tensorflow/issues/1727). Since the Python ML module needs to remain active for the hybrid model, memory cannot be freed and will continue to accumulate over time.*

Lines 373-383: for provenance it could be useful to see the code used to perform these tests.  As far as I can tell it is not included in the Zenodo archive.

*Reply: Thanks for the suggestion. We have included the code at https://github.com/tzhang-ccs/ML4ESM/tree/main/cffi2cython.*

Figure 11: it is sort of surprising that the single column model is slower than the ne4 configuration.  Is there not a way to get it to run faster than ne4?

*Reply: This is because NE4 uses 128 cores for parallel computation, whereas SCM only uses a single core.*

Data Availability Statement: I understand the long-term value of storing the code in a Zenodo archive, but could you also include a link to the code on GitHub?  This makes it easier for people to quickly read and review, rather than downloading and unpacking the code from Zenodo.

*Reply: Thanks for the suggestion. The codes are available at [https://github.com/tzhang-ccs/ML4ESM](https://github.com/tzhang-ccs/ML4ESM).*

Technical corrections

Line 217: "applied it two" -> "applied it in two"

*Reply: revised*

Line 218: "CAPE-based trigger function in deep convection" -> "CAPE-based trigger function in a deep convection"

*Reply: revised*

Line 219: "machine-learnt" -> "machine-learned"

*Reply: revised*

Line 404: "A same" -> "The same"

*Reply: revised*

Line 427: "Compassion CNTL" -> "Comparison of CNTL"

*Reply: revised*

References

Kochkov, D., Yuval, J., Langmore, I., Norgaard, P., Smith, J., Mooers, G., Klöwer, M., Lottes, J., Rasp, S., Düben, P., Hatfield, S., Battaglia, P., Sanchez-Gonzalez, A., Willson, M., Brenner, M. P., & Hoyer, S. (2024). Neural general circulation models for weather and climate. Nature, 632(8027), 1060–1066. https://doi.org/10.1038/s41586-024-07744-y.

Rasp, S., Pritchard, M. S., & Gentine, P. (2018). Deep learning to represent subgrid processes in climate models. Proceedings of the National Academy of Sciences, 115(39), 9684–9689. https://doi.org/10.1073/pnas.1810286115.

---

## Author Response (AR2)

**Summary**

I appreciate the authors' revisions and responses to both my and the other reviewer's comments. I think the structure and focus of the manuscript have improved since the initial submission. I have a few remaining minor comments / suggestions, but overall I think this manuscript can be accepted subject to minor revisions.

**Response to responses**

> Reply: Sorry for the confusion. Each bar in Figure 10 represents the performance for the hybrid system that couples the ML methods into the ESM, not just the ML module alone [….].

Indeed I understood this, but was wondering about how relevant this discussion was in a paper focused on a new coupling approach. Nevertheless, I think what is in the manuscript is OK. It is useful as a case study on the impact of size of representative ML model on computational performance relative to a representative GCM, and is facilitated by the flexibility of the coupling approach. I agree it is also useful as a point of reference to compare to previous work, assuming it is roughly an apples-to-apples comparison.

Reply: Thank you for your comments. In Wang et al. (2022), a ResNet deep learning model was used. To enable a fair comparison, we also tested the overhead of ResNet in our study. This allows for a direct comparison of the computational overhead associated with different coupling approaches.

> Reply: This is because NE4 uses 128 cores for parallel computation, whereas SCM only uses a single core.

Would it be possible to increase the single column model performance by running with more cores? Again it is surprising that a simulation with multiple interacting columns is faster than a simulation with a single column, but maybe that is just due to differences in the typical processor layouts (no need to make any changes).
Reply: Yes, it is possible to use more cores for a single column model by decomposing the vertical layers. However, most climate models, including E3SM, use 2D horizontal decomposition. For a single column model, the computational cost is significantly lower than that of a global model, so vertical decomposition is typically unnecessary.

I understand your concern that the NE4 configuration might appear faster than the single column model, even though it uses a very coarse resolution and more cores. The reason could be that the single column model involves additional overhead, such as the higher frequency of output.

**Specific comments**

Lines 76-79: "The hard-coding has limitations. Such hard-coding approach restricts the ML algorithm's ability to adapt to changes in the model dynamics over time, as the 'online' updating requires a two-way coupling between the dominantly Fortran-based ESMs and Python ML libraries." I am still not sure I fully understand what is meant by "two-way coupling" in this context. Other than flexibility, how is the coupling used in O'Gorman and Dwyer (2018) conceptually different than that offered by FKB, CFFI, or the authors' new approach? In all of those cases, the ML model is typically trained offline, and is in a sense frozen when coupled online—like any parameterization, the only way its outputs change is as a result of its inputs changing. Are the authors envisioning some sort of "online" training approach that does not depend on the differentiability of the GCM?

Replay: In this study, the term 'two-way' means that the offline-trained ML model not only affects the GCM when coupled to it but also allows the GCM, particularly a potentially differentiable dynamical core, such as NeuralGCM, to 'online' update the trained ML model. O'Gorman and Dwyer (2018) saved the trained ML model in a NetCDF file. Then the GCM loads it. In their approach, the ML model is frozen and can not update during runtime.

In contrast, methods such as FKB, CFFI, and our approach establish an interface between Fortran and Python, where the ML model is loaded on the Python side. This design provides the flexibility to update the ML model with a differentiable dynamical core during runtime, which enables the possibility of online learning.

We have added the following discussion in the manuscript "When a trained ML model is incorporated into ESMs, it is frozen and cannot be updated during runtime. Recently, Kochkov et al.(2024) introduced the NeuralGCM, an innovative approach that enables the ML model to be updated during runtime with a differentiable dynamical core. This allows for end-to-end training and optimization of the interactions with large-scale dynamics. However, the hard-coding coupling method does not support such capability. "

Figure 2: this is great—thanks for adding it, as well as the details about compilation. Maybe mention in the caption that a fleshed-out, compilable version of this toy example exists in the linked GitHub repository as well.
Replay: Thanks for the suggestion. We have added this in the caption.

Lines 512-513: "In contrast, Wang et al. (2022) reported a 100% overhead in their interface, which transfers parameters via files." What kind of model was used in Wang et al. (2022)? Is it comparable to the one tested here?
Replay: In Wang et al. (2022), a ResNet model was used to train the ML parameterization. To ensure a fair comparison, we also tested the ResNet ML model in our study. We have clarified this in the manuscript.

Line 519: "predicts the computational ratio relative to the CNTL run by taking the number of ML parameters as input" is somewhat vague. I might suggest using something like: "predicts the

ratio of the simulated years per day of the ML-augmented run to that of the CNTL run as a function of the number of ML parameters"
Replay: Thanks for the suggestion. Revised it.

Figure 9b: this is very minor, but it might make things a little more intuitive to read if the y-scale for the ratio exactly corresponded with the y-scale for the simulated years per day (i.e. ran from something like 0.222 to 1.111). This way the "truth" line would run through the top of each bar.
Replay: Thanks for the suggestion. Revised it.

**Technical corrections**

Line 132 "only minimal disrupting" -> "only minimal disruption to"
Replay: Revised it.

Line 161: "using toy code example" -> "using a toy code example"
Replay: Revised it.

Line 169: "real model" -> "fortran model"
Replay: Revised it.

Line 276: "[...] (1.5 km grid spacing). Met Office [...]" -> "[...] (1.5 km grid spacing) Met Office [...]" (i.e. remove the period).
Replay: Revised it.

Figure 10 caption: "Compassion" -> "Comparison"
Replay: Revised it.